# Information Fragmentation, Encryption and Information Flow in Complex Biological Networks

**DOI:** 10.3390/e24050735

**Published:** 2022-05-21

**Authors:** Clifford Bohm, Douglas Kirkpatrick, Victoria Cao, Christoph Adami

**Affiliations:** 1Department of Integrative Biology, Michigan State University, East Lansing, MI 48823, USA; 2BEACON Center for the Study of Evolution in Action, Michigan State University, East Lansing, MI 48823, USA; dkinventor@gmail.com (D.K.); caovicto@msu.edu (V.C.); 3Department of Computer Science and Engineering, Michigan State University, East Lansing, MI 48823, USA; 4Department of Microbiology and Molecular Genetics, Michigan State University, East Lansing, MI 48823, USA; 5Program in Ecology, Evolution, and Behavior, Michigan State University, East Lansing, MI 48824, USA

**Keywords:** information processing, information fragmentation, neural network evolution, computational complexity

## Abstract

Assessing where and how information is stored in biological networks (such as neuronal and genetic networks) is a central task both in neuroscience and in molecular genetics, but most available tools focus on the network’s structure as opposed to its function. Here, we introduce a new information-theoretic tool—*information fragmentation analysis*—that, given full phenotypic data, allows us to localize information in complex networks, determine how fragmented (across multiple nodes of the network) the information is, and assess the level of encryption of that information. Using information fragmentation matrices we can also create information flow graphs that illustrate how information propagates through these networks. We illustrate the use of this tool by analyzing how artificial brains that evolved in silico solve particular tasks, and show how information fragmentation analysis provides deeper insights into how these brains process information and “think”. The measures of information fragmentation and encryption that result from our methods also quantify complexity of information processing in these networks and how this processing complexity differs between primary exposure to sensory data (early in the lifetime) and later routine processing.

## 1. Introduction

Networks such as brains and gene regulatory networks generate actions by operating on sensory information in order to implement behaviors or phenotypes, such as firing muscle neurons or activating a particular set of genes. The simplest networks generate responses directly from the current state of individual sensors, or perhaps integrate information from multiple sensors, to generate reflexive behaviors. More complex networks may implement memory (the storage and later retrieval of information) that can be integrated with current sensory data to produce more informed behaviors dependent on prior states, such as neural adaptation, habituation, and learning.

How these networks compute, process, and integrate information from various sources has been the subject of intense research. However, when information is encoded in multiple units, it has proven difficult to link network activation states (firing or activation patterns) to behavior or function, for example to predict the state of “downstream” decision neurons (or genes) armed only with the state of “upstream” neurons or genes. In this context, functional connectivity research [1] seeks to characterize statistical associations between neural time series and behavior, but disambiguating causation from correlation has been plagued by confounding variables. In genetics, genome-wide association studies (GWAS) attempt to link genetic variants with traits (typically diseases), but the focus on single-nucleotide polymorphisms has led to weak signals and poor predictive performance [2]. While information-theoretic methods have been developed to characterize the information transmission capacity of single neurons [3] and to study how information is used in decision making [4], again the focus there is on prediction based on information encoded in single units. There is a body of theory that attempts to extract information stored in multiple variables (see, e.g., [5,6]) but these constructions rely on an information decomposition that makes use of a definition of “specific information” that is questionable because it is not additive [7].

Here, we outline a formalism that allows us to find information that is encoded in multiple variables and is predictive of arbitrary traits (that we specify), using standard Shannon information theory. Because this theory relies explicitly on the availability of the *joint* state of all variables in the system, it is necessarily limited in scope because the joint state of variables is often not made available. In neuroscience, usually a few time series of individual neurons are given (along with the time series of behaviors), while in GWAS the effect of single nucleotide substitutions is probed. The work most similar to the present effort that we are aware of is “reconstructability analysis”, which seeks to find minimal multivariate models that explain certain data [8].

While the tools we introduce can be applied to any information-bearing system of nodes (even in the absence of structure information such as physical links between nodes), here we test those tools to investigate decision-making models trained via digital evolution. These models result in relatively simple structures from which complete phenotypic data can be obtained and analyzed. In particular, we use two artificial brain models: Markov Brains that use arbitrary logic gates and sparse wiring; and recurrent neural networks (RNNs), which are fully connected networks of summation and threshold nodes. We have identified other systems that we could apply our tools to, such as simulated genetic networks (for example, the *Drosophila* segment polarity network [9]) but this is left for another publication.

### Entropy and Information

Information is not additive and need not be localized. While these facts may be obvious to information theory practitioners, they may be less intuitive to those that use the word “information” colloquially. We begin with a (very brief) overview of entropy and information, the central concepts needed to understand the tools presented in this article. For a more in-depth explanation of what information is, *and what it is not*, see [10].

Information, as defined by Shannon, allows the holder to make predictions about a system with accuracy better than chance [11]. In order to talk about information, we must first define the units of information, and thus it is useful to first define entropy. For a discrete random variable *X* that can take on the states $x_{1},\cdots ,x_{n}$ with probabilities $P\left(X=x_{1}\right)=p_{1},P\left(X=x_{2}\right)=p_{2},\cdots ,P\left(X=x_{n}\right)=p_{n}$ the Shannon *entropy* of *X* is defined as
(1)$$\begin{array}{c} H\left(X\right)=-\sum_{i=1}^{n}p_{i}\log p_{i}\;. \end{array}$$

Entropy characterizes how much we do *not* know about *X*, and the base of the logarithm gives units to the entropy. Low entropy tends to be associated with systems that have few states, while high entropy tends to be associated with systems with numerous states.

Information is a measure of how much the entropy in one system is correlated with the entropy of another system. Intuitively, a coin toss has 1 bit of entropy (either heads or tails with equal probability). Note that whenever we flip a coin, the hidden side of the coin also has one bit of entropy. Viewed as two independent variables, the coin-flipping system has two bits of entropy. However, since the face-up and face-down values are perfectly correlated, the combined system only has one bit of entropy: the face-up value provides information about the face-down value. We denote information (*shared entropy*), between the variables *X* and *Y* with I(X;Y). The semicolon between *X* and *Y* in the shared entropy is to remind us that information is symmetric between the systems: what *X* knows about *Y* is precisely what *Y* knows about *X*.

In order to calculate the entropy shared among a set of variables, we must first introduce the idea of joint entropy. When we calculate entropy, we do so by counting the number of times each unique state appears in a system to determine the fraction of time the system is in each state (each $x_{i}$) and using Equation (Equation 1). We can calculate a shared entropy by simply treating two or more systems as though they were a single system. Consider a system with two binary variables *A* and *B*. We could observe these two variables independently and each would generate a list of states (0s and 1s), or we could also observe *A* and *B* jointly, which would generate a list of states where the states are pairs of bits (which could alternatively be represented as a list of values from the set {0,1,2,3}). The joint entropy is obtained using the standard entropy calculation on the joint states.

The formula for joint entropy using the variables *X* and *Y* can be written as (here, XY is the joint variable constructed from *X* and *Y*)
(2)$$\begin{array}{c} H\left(XY\right)=-\sum_{x\in X}\sum_{y\in Y}p\left(x,y\right)\log\left[p\left(x,y\right)\right]\;, \end{array}$$
where p(x,y)=P(X=x,Y=y) is the joint probability to find *X* in state *x* while at the same time *Y* is in state *y*. The entropy of a joint system is at least as large as the single largest independent entropy (adding to a system cannot simplify that system). Moreover, the shared entropy of a joint system cannot be greater than the sum of all of its parts, and that maximum is only attainable if all the individual variables are uncorrelated.

Finally, information is the difference between the sum of a set of independent-variable entropies (so-called “marginal” entropies) and the joint entropy of the set:(3)$$\begin{array}{c} I(X;Y)=(H(X)+H(Y))-H(XY)\;. \end{array}$$
The non-additivity of information is easily seen by studying an *X* that has two parts: $X=X_{1}X_{2}$ (a joint random variable), and asking how much knowing *Y* tells us about the two parts of *X*. As described above, entropy is non-additive:(4)$$\begin{array}{c} H\left(X\right)=H\left(X_{1}X_{2}\right)=H\left(X_{1}\right)+H\left(X_{2}\right)-I\left(X_{1};X_{2}\right)\leq H\left(X_{1}\right)+H\left(X_{2}\right)\;. \end{array}$$
So we see that the entropy of *X* is not just given by the sum of its constituent entropies, because those parts can *share* entropy. Here, the joint entropy $H(X_{1}X_{2})$ is strictly smaller than (or equal to) the sum of the constituent entropies. This carries over to informatios, as
(5)$$\begin{array}{c} I\left(X;Y\right)=I\left(X_{1}X_{2};Y\right)=I\left(X_{1};Y\right)+I\left(X_{2};Y\right)-I\left(X_{1};X_{2};Y\right)\;. \end{array}$$
How the information in *X* is spread out into information in $X_{1}$ and in $X_{2}$ is illustrated in the entropy Venn diagram shown in Figure 1.

Equation (Equation 5) makes it clear that information is not only non-additive; Equation (Equation 5) also demonstrates that information can be distributed (fragmented) across multiple systems (in this example, only two). Some authors have attempted to separate the “synergistic” and “redundant” parts of the information shared between three or more variables ( [5,12], see in particular the overview [6]) using an information decomposition, but these concepts rely on a definition of “specific information” that is constructed to avoid negative values, which has a serious flaw [7]. The negativity of terms such as $I(X_{1};X_{2};Y)$ (sometimes called “multi-information” [13]) can be intuitively understood in terms of cryptography, as we will see later on.

In the following, we will develop tools that find and isolate subsystems $X_{i}$ that are a subset of a whole *X* and harbor information about some *Y*.

## 2. Information Fragmentation

To define information fragmentation, we first define the concept of a *distinct informative set*, or DIS. For a variable *X* with *m* elements (or parts) $X_{i}$, we can define the sets $S_{m}$ of *X*, which are all the different ways in which elements of *X* can be combined (without repetition): the power set. So for example, for a variable with three elements, the power set is:(6)$$\begin{array}{c} S_{3}=\left\{\emptyset ,\left\{X_{1}\right\},\left\{X_{2}\right\},\left\{X_{3}\right\},\left\{X_{1},X_{2}\right\},\left\{X_{1},X_{3}\right\},\left\{X_{2},X_{3}\right\},\left\{X_{1},X_{2},X_{3}\right\}\right\}\;. \end{array}$$
The number of possible sets of a variable with *m* parts (the cardinality of the set $S_{m}$) is $2^{m}$, which makes an exhaustive search across all elements of the set impractical for large systems, but we shall see that often an exhaustive search is not necessary to calculate fragmentation.

An informative set $S_{X}$ is an element of $S_{m}$ that contains non-zero information about *Y* (a variable of interest):(7)$$\begin{array}{c} I(S_{X};Y)\geq 0\;. \end{array}$$
We then define a *distinct informative set* (DIS) $S_{X}^{⋆}\left(\theta \right)$ as any irreducible informative set that has at least θH(Y) information about *Y* (with 0≤θ≤1):(8)$$\begin{array}{c} I(S_{X}^{⋆}\left(\theta \right);Y)\geq \theta H\left(Y\right)\;. \end{array}$$
Here, an irreducible set is a set that is not a proper superset of another set with $I(S_{X}\left(\theta \right);Y)\geq \theta H\left(Y\right)$. In other words, a set is “distinct” only if it *does not contain* a smaller predictive set. It is possible that a feature is predicted by more than one DIS: a *set of distinct informative sets*, or DISs. The parameter θ represents the fraction of the entropy of *Y* that we would like to explain, so that if we want to find DISs that explain *Y* perfectly (that is, predicts the state of *Y* with perfect accuracy), then θ=1.

When more than one DIS can predict the same feature, we call such DISs *redundant predictive sets*. A situation may arise when analyzing a network with causal relationships (e.g., a network where prior states are known to affect later states), where one DIS is actually causal while the other(s) simply mirrors the causal state, but is not itself causal; i.e., it is a *coincidental prediction*. If we have access to the internal connections of the network (i.e., the connectome in our digital brains), we are able to resolve some cases of ambiguity by analyzing the connections in order to identify cases of coincidental prediction where, at time t−1, a particular set predicts some time *t* feature, but there are no connections in the network that can account for causation. Another more comprehensive process to resolve coincidental prediction is described in the Methods.

The quantity we call *fragmentation* $F_{\theta }\left(X;Y\right)$ of a *feature Y* (a feature is a variable of interest that we would like to predict) is then simply the *size* of the smallest DIS; that is,
(9)$$\begin{array}{c} F_{\theta }\left(X;Y\right)=\min_{S_{X}^{⋆}}\left|S_{X}^{⋆}\left(\theta \right)\right| \end{array}$$
measures the degree of fragmentation of the information (about *Y*) that is stored in a distributed system $X=X_{1}\cdots X_{m}$. Note that it is possible that there is no part of *X* that can account for some θ worth of information in *Y*. In this case, the fragmentation is assigned -1 to indicate that no valid set of elements of *X* was found.

$F_{\theta }$ can be obtained by complete enumeration of all the shared entropies between *Y* and $S_{X}\in S_{m}$. However, such an exhaustive search is only necessary when no subset of *X* that contains θ information about *Y* is found, because we can stop searching as soon as we find a sufficient subset of *X*. For this reason, we conduct our search of the possible sets of $S_{m}$ from smallest to largest.

An extension of fragmentation analysis involves calculating all the shared entropies between all subsets of *X* and some *Y* in order to generate an *information fragmentation matrix*. Here, *Y* may be a list of features such that if *Y* is composed of *k* features (e.g., neurons or traits) $Y=Y_{1}\cdots Y_{k}$, then the fragmentation matrix F has elements
(10)$$\begin{array}{c} F_{ik}=I\left(S_{X}^{\left(i\right)};Y_{k}\right)\;, \end{array}$$
where $S_{X}^{\left(i\right)}\in S_{m}$ is an element of the power set of explanatory variables *X* while $Y_{k}$ is a variable to be explained. It is often convenient to normalize the matrix elements by the entropy of the feature to be explained, in which case the normalized fragmentation matrix is defined as
(11)$$\begin{array}{c} {\hat{F}}_{ik}=\frac{I(S_{X}^{\left(i\right)};Y_{k})}{H(Y_{k})}\;. \end{array}$$

The fragmentation matrix reveals what set of variables of *X* correlate with (and therefore predict) which features of *Y*. A predicted feature could be a statement about the real world, such as, “Which city has a larger population, Hamburg or Cologne?” (the *German-City Problem*, see [14]). In this problem, the predictive variables *X* are cues such as “Has a subway system”, “Is the national capital”, or “Has a university”.

A fragmentation matrix can also be used to track the flow of information in a complex dynamical system. In neuronal networks, the set *X* can be the firing state of neurons at time *t* while *Y* could be the same set of neurons at time t+1. Or, in genetic systems, *X* can be the activation state of a set of pair-rule genes that determines a developmental process (e.g., the patterning of the syncytial blastoderm in *Drosophila*), and *Y* is the same set of genes at a later time point.

Here, we study information fragmentation by evolving digital artificial brains using two different substrates. The first substrate, “Markov Brains” [15], is built from sparsely connected arbitrary logic gates (think of evolvable computer circuits). The second type, “recurrent neural networks” (RNNs), is built from standard continuous-value Hopfield-type neurons with a sigmoid transfer function. Both brain types include recurrent connections to allow for memory.

We consider two different tasks. The first is a memorization task: the object is to read back a set of inputs with different delays (a task we call “*n*-Back” [16]). The second task is an active perception task (as pioneered by Beer [17,18]), where the agent has to make a decision on whether to catch or avoid a falling block based on its size and/or direction of motion. The task is difficult because the block has to be perceived over time (actively) to make optimal decisions. The tasks, along with details about the implementation of the neural substrates, are described in more detail in the Methods section.

## 3. Results

We study information fragmentation within digital brains evolved in different environments that reward different tasks. The brains that we are using—Markov Brains and Recurrent Neural Networks (RNN)—have identical input, output and memory systems, while the number of input nodes and output nodes are task dependent and the number of memory nodes is fixed at eight. The brains differ in their logical operation units, how those units are interconnected, and in encoding schemes (see Methods for a more complete description of the brains). For the majority of our analysis using fragmentation matrices and information flow, we will use Markov Brain results. In the last experiment, we also use RNN results to provide additional support for our claims.

### 3.1. n-Back Task

The first task is *n*-Back [16], a simple memory problem where an agent is provided a sequence of binary digits through a single sensory node, one at a time. The inputs must be stored in memory and read out with particular delays. Here, the agent has five outputs labelled $o_{1}$ to $o_{5}$, which (for perfect performance) should display the values at time *t* that were previously seen at time t−1, t−3, t−5, t−7, and t−8. The second task, Block Catch [17,18], is a more complex task, requiring agents to classify blocks by size and direction of movement and then execute behavior to catch or avoid blocks based on the classification. In Block Catch, agents receive inputs through four input nodes (a visual sensor array with a gap in the middle) and use two output nodes to activate motors that allow for left and right movement. Figure 2 shows the input, output, and memory node configuration for each task. See Methods for a more complete description of the tasks.

For Markov Brains, echoing the input from t−1 is the easiest aspect of this task and consequently evolves first [16]. Echoing values with longer delays is progressively more difficult, and builds on the computational architecture that has previously evolved. Correctly outputting a bit that was read eight time-steps earlier (t−8) is the hardest to achieve, and evolves last. Across 250 evolutionary runs, each run for 40,000 generations, 100 Markov Brains achieved perfect performance.

We show in Figure 3a–d parts of four different Markov Brain fragmentation matrices that all evolved perfect performance. These four represent a range of complexities starting with (a): a brain with the simplest possible solution, and progressing through (b), (c), and (d), which each represent increasingly complex solutions. The matrix is organized in such a way that the predicted variables (here, the five outputs $o_{1}$ to $o_{5}$) label the rows of the matrix, and the set of predictors (the power set constructed from the eight memory nodes $m_{1}$ to $m_{8}$, excluding the empty set) labels the columns. For this task, the fragmentation matrix is 5×255. The value of the matrix element is the fraction of available information the set $S_{j}$ has about the *expected* outputs at the next time point, that is, $F_{ij}=I\left(o_{i}\left(t+1\right);S_{j}\left(t\right)\right)$. When displaying the elements of the fragmentation matrix in Figure 3, the gray scale indicates how close an element is to perfect prediction (as the inputs and therefore the outputs are chosen to be random bits, the entropy of a perfect prediction in this matrix is 1 bit).

The matrix in Figure 3a shows that the single node $m_{4}$ predicts output $o_{1}$; therefore, $m_{4}$ is a DIS for the feature $o_{1}$. For this brain, *all* outputs are predicted by single nodes, which means that information is not fragmented (i.e., each feature has fragmentation 1). The same is true for the fragmentation matrix in Figure 3b, except of course that different nodes are correlated with different outputs, as these brains are evolved independently.

The matrix shown in Figure 3c has a single feature (output $o_{5}$) that is predicted by a pair of nodes, the set {$m_{2},m_{7}$}. This set is the smallest DIS for this feature, so the feature has fragmentation 2. Furthermore, it is clear that neurons $m_{2}$ and $m_{7}$ separately do not predict $o_{5}$ at all. This is a common occurrence in evolved Markov Brains and informs us that information is distributed among those two nodes in a *cryptographic* manner (technically, a Vernam, or one-time pad, cipher [19]), as shown in Figure 4. In the figure, each circle (the entropy of each node $m_{2}$, $m_{7}$, and $o_{5}$), contains four values that sum to 1 (0+1+1+−1=1), indicating that each node’s entropy is 1 bit. Moreover, the shared entropy between each pair of nodes is 0 bits while the joint entropy of each pair of nodes is 2. As a consequence, when viewed two-at-a-time, each pair of variables appears to be uncorrelated. Only when we consider all three nodes at the same time are we aware that the total system entropy is 2, not 3. In a sense, each node serves as the cryptographic key that unlocks the information stored between the other two. Without the key, the information is completely hidden: the hallmark signature of one-time-pad encryption. Note that the negativity of the multi-information is also sometimes called “synergy” [6].

Figure 3d depicts the fragmentation matrix for a brain in which information is more fragmented. While outputs $o_{1},o_{2},o_{4}$ and $o_{5}$ are predicted by single neurons, the smallest DIS for $o_{3}$ is the tripartite set $\left\{m_{1},m_{3},m_{8}\right\}$. Note that in all the matrices, the joint set $\left\{m_{1},\cdots ,m_{8}\right\}$ (the right-most set in Figure 3a–d) predicts all outputs perfectly, indicating that each output’s information is contained in the brain when viewed as a whole.

Using a fragmentation matrix and the concept of DISs, we can follow the flow of information as the digital brain (or any causal network of nodes) performs computations. Measuring information flow through a network is a difficult task in principle when the time-ordering of activations of nodes is unknown. This is typically the case if only the recordings of individual nodes is given, rather than the joint state of all nodes. In such a case, it is typically necessary to use measures such as Granger causality [20] or transfer entropy [21] (see also [22,23,24,25]). When the joint state of all nodes is given, the task is significantly easier as the time-ordering of node activations is known.

To reconstruct information flow, we first generate a new normalized fragmentation matrix using a *different* set of features (the output and memory nodes at time *t*) and predictors (the input and memory nodes at time t−1) and θ=1 (i.e., perfect prediction). These matrices are large; 9 (the number of features) by 8192 (the size of a power set with 13 elements), and so are not shown. We use the normalized fragmentation matrix to reconstruct the causal chain that results in current values of particular outputs $o_{i}$ and memory $m_{i}$ at time *t*, by searching the fragmentation matrix for DISs that, at time t−1, predict each time *t* state. In a complete system (that is, one where the state of all the nodes are accessible to us), we can always identify all the DISs $S_{j}\left(t-1\right)$ that perfectly predict each feature. So, the nodes at t−1 that predict the state of some node $n_{i}\left(t\right)$ (one of the nodes $o_{i}\left(t\right)$ or $m_{i}\left(t\right)$) are identified when we find the sets $S_{j}\left(t-1\right)$ for which
(12)$$\begin{array}{c} I\left(S_{j}\left(t-1\right);n_{i}\left(t\right)\right)=H\left(n_{i}\right)\;. \end{array}$$
Once the sets of nodes at t−1 are found that predict $n_{i}\left(t\right)$, we draw an arrow from *each of the elements* of these sets $S_{j}\left(t-1\right)$ to $n_{i}\left(t\right)$, indicating information flow. The arrows are colored black unless we cannot determine that an arrow is necessary—as is the case when we can not resolve redundant predictive sets—in which case the arrow is colored red. In addition, we label each node with the entropy recorded in that node, and we label each arrow with a value indicating the proportion of the downstream node’s entropy that is accounted for by the upstream node.

In Figure 5, we show flow diagrams for the four brains whose fragmentation matrices are shown in Figure 3. The flow diagrams for each of the four brains, labeled (a) through (d), show increasingly fragmented information, where more complex information flow appears as graphs with more edges between nodes. We thus define “information flow complexity” simply as the total number of edges in an information flow graph, and conjecture that this flow complexity is a proxy for computational complexity. Note that when a node is predicted by only one DIS, the in-degree of the node is the fragmentation F of the node.

For the simplest solution shown in Figure 5a, information flow is straightforward. Bits that are read in $i_{1}$ are simply passed from one node to the next, without any fragmentation.

In the brain whose fragmentation matrix was shown in Figure 3c, the output node $o_{5}$ was predicted by the nodes $m_{2}$ and $m_{7}$, but each of the nodes did not convey any information independently, indicated by the number 0 next to each arrow. Moreover, $m_{7}$ at time *t* is predicted by $\left\{m_{2},m_{6}\right\}$ at t−1, while $m_{2}$ at *t* is predicted by $m_{6}$ at t−1. Taken together, this analysis reveals the algorithm that ensures that $o_{5}$ carries the signal two time steps removed from the signal in $o_{3}$ (as the task requires). Node $m_{6}$ encodes the signal at time *t* (call it $x_{t}$). This value is copied into node $m_{2}$, which is then combined in a logical XOR (⊗) with the previous value, that is, $m_{7}$ carries $x_{t}⊗x_{t+1}$. In the next step of the algorithm, $m_{7}$ is combined again with $m_{2}$ (which carries $x_{t+1}$) in an XOR to determine $o_{5}$:(13)$$\begin{array}{c} o_{5}\left(t+3\right)=\left(x_{t}⊗x_{t+1}\right)⊗x_{t+1}=x_{t}⊗\left(x_{t+1}⊗x_{t+1}\right)=x_{t}⊗0=x_{t}\;. \end{array}$$
Thus, through consecutive encryption (really an encryption/decryption cycle) with the same key (here $x_{t+1}$), the brain achieves a simple time delay. While cumbersome, this method of time delay is not rare in our results. We can see encryption in Figure 3b, where the internal node $m_{3}$ is the center of an encryption/decryption cycle involving $m_{1}$, $m_{5}$, and $m_{8}$; even though *none* of the features $o_{1}-o_{5}$ in that brain are fragmented. In Figure 3c, two similar encryption cycles are present; resulting in $m_{3}$, $m_{7}$, $m_{8}$, and $o_{5}$ having fragmentation F=2. The flow diagram Figure 5d is even more complex, where $m_{1}$ has F=2 and nodes $m_{5}$, $m_{6}$, $m_{8}$, and $o_{3}$ have fragmentation F=3, indicating pervasive encryption throughout the network.

### 3.2. Block Catch Task

The Block Catch task is an “active perception” problem where an agent has to make a decision to either catch or avoid falling blocks of different sizes with either left or right lateral motion, based on the size and lateral movement direction. The sensors have blind spots designed so that a perfect agent must integrate multiple sensor readings over time to determine block size and lateral motion [26,27]. A schematic of the task can be seen in Figure 6b and the details of the task are described in the Methods section.

Figure 7a–c show parts of the fragmentation matrices from three brains evolved on the Block Catch task that achieve perfect fitness using Markov Brains. Here, the features to be predicted (rows) are neither expected outputs nor node states, but rather states of the world that we expect would be good to know in order to perform well in this world. For example, the first row of the matrices in Figure 7 encodes information about whether a block is to be caught or not, while the second row encodes information about the block’s lateral movement. Rows 3–6 reflect potential information about block size (we included block size 1 as a control: as such blocks are never used, that row should remain dark). The last four rows display information about combined concepts, such as “Is the block moving left and to be caught?”. For each brain, we show two matrices, one showing fragmentation values for “full lifetime” and the other for “late lifetime” (the last 25% of each block’s trajectory). If information is built up during an agent’s lifetime, we expect to see higher values of prediction in the late-lifetime matrices, which are collected from the period during which the brain should have already “made up its mind”.

Because the Block Catch task requires agents to determine the block size and lateral direction during their lifetime, and because this process requires the agent to observe the blocks for some number of updates, we do not expect to see perfect information about any features in the fragmentation matrices for full lifetimes. Interestingly, even when we look at the late-lifetime matrices, we do not see perfect information about many of the features. A visual inspection of the evolved behavior (not shown) suggests that, indeed, those brains have (at least for the most part) already “made up their mind” about the required behavior prior to the period covered by the late-lifetime matrices. From the fragmentation matrices, we must conjecture that the brains are using strategies that do not require localization of world-state information in the brain. But what could account for perfect performance while not having perfect late-lifetime information?

Since an agent can affect the state of inputs by implementing a particular behavior (can “choose what they will see”), one possibility is that the agent’s movements result in an input pattern that “offloads” information [28,29], that is, the agent can extend their memory by affecting the world state and access this memory through the sensors. Another possibility is that the brains perform their classification, choose a behavior and lock it in, after which they “forget” the world-state information that was used to make the decision. Regardless, in cases where information is built up during the agent’s lifetime, we should see higher values of prediction in the late-lifetime matrices. In the three brains that we present (and in others we analyzed) we see that there are increases in information prediction later in life. In Figure 7(b.2) we can see that $m_{3}$ predicts direction, for example. On the other hand, it is clear that the brains are not required to have very much localized information to perform the Block Catch task, as we now document.

The information flow for the Block Catch task is visualized in the same way as described earlier for the *n*-Back task (see also Methods). In Figure 8 we see that the full-lifetime flow diagrams are more complex than the late-lifetime flow diagrams, indicating that the nature of computations changes significantly between early-lifetime and late-lifetime. To be clear, these plots do not indicate that the structure of the brains is changing over lifetimes; what is changing is only which elements and connections in the brain are being used and how information is moving though these elements and connections. Early in life (i.e., in the first 75% of lifetimes), the agent must collect and integrate sensor data to ascertain the size and lateral motion of a block, which must be further integrated to activate motors that enact behavior resulting in either catching or avoiding the block. During the last quarter of the lifetime, this decision is often already made, and the brain simply follows through on this decision. This is clearly visible in the simplicity of the late-lifetime information flow diagram in Figure 8(a.1).

By way of illustration, let us consider the entropy of the inputs in the late-lifetime flow diagram, Figure 8(a.2). Remember that all size-2 blocks and size-3 blocks with right lateral motion should be caught. An inspection of the behavior of this brain shows that for size-2 blocks, the agent positions itself so that the block falls between the left and right pairs of sensors, while for right-moving size-3 blocks the agent positions itself so that only the right center sensor ($i_{3}$) is covered. For all other blocks (i.e., the blocks that should be avoided) all sensors are off in late lifetime. As a result, the sensor nodes $i_{1}$,$i_{2}$, and $i_{4}$ have zero entropy, (they are never on) during late life. There are six block types in total, so $\frac{1}{6}$ of the time $i_{3}$ is on and the other $\frac{5}{6}$ of the time $i_{3}$ is off. By applying the equation for entropy (Equation (Equation 1)), we see that the late-lifetime entropy for $i_{3}$ is 0.65.

The lack of arrows (in late lifetime information flow diagram, Figure 8(a.2)) from any of the inputs indicates that the inputs provide no information about late-life behavior. On the other hand, the late-lifetime flow diagram clearly shows that movement information is stored in memory node $m_{3}$, which sets the output (i.e., motor) nodes accordingly. The fragmentation matrix Figure 7(a.2) for this brain indicates that at this point in time, the early-life behavior has correctly positioned the agent relative to the current block, and the only thing that the brain knows for sure is what direction to move in (i.e., if the brain at some point knew the block size, it has now forgotten it).

The medium complexity brain (fragmentation matrices in Figure 7(b.1,b.2), flow diagram in Figure 8(b.1,b.2)) has a significantly more complex information flow compared to the simple brain. That agent is able to make its decision without the aid of sensor $i_{4}$, but uses six memory nodes supporting complex information flow, as compared with the simpler brain (a) which uses all four sensors, but only two memory nodes. As in the simple brain (a), the late-lifetime computation is less complex, but this brain still pays attention to the sensors during the late-lifetime period.

The third (most complex) brain (shown in Figure 7(c.1,c.2) and Figure 8(c.1,c.2)) uses eight internal nodes, with information processed through as many as five time steps before reaching the motors from the sensors. While some aspects of the information dynamics can be gleaned from the information flow over the lifetime, it is not sufficient to infer an algorithm, in particular because the computation changes as time goes on. The late-lifetime information flow diagram for this brain (shown in Figure 7(c.2)) follows the trend of the other Block Catch brains and is simpler than the full-lifetime flow. For example, signals go either directly from sensors to motors, or else have only a single internal node in between. Moreover, where there were feedback loops in the full-lifetime diagram (i.e., between $m_{2}$ and $m_{4}$) information flows only forward (downstream) in the late-lifetime diagram. Finally, in the late-lifetime diagram for this brain, we see cases of *redundant predictive sets* (multiple DISs predicting the same feature): both $m_{4}$ and $o_{2}$ can be predicted by two distinct sets of nodes. For example, $m_{4}$ is predicted by either the set $\left\{i_{2},m_{4},i_{4}\right\}$ or the set $\left\{i_{2},m_{4},m_{7}\right\}$. Since $i_{2}$ and $m_{4}$ are present in both sets, we know that they must be necessary but, we do not know whether it is the set containing $m_{7}$ or the set containing $i_{4}$ that is responsible for $m_{4}$ (which is why we colored those two arrows red). Likewise, the output $o_{2}$ is definitely predicted in part by $i_{1}$, but either $m_{7}$ or $i_{4}$ are part of the set that actually determines this node. Note that in this case these redundancies could not be resolved by examining the brain’s connectome, which indicates that there were physical pathways in the brain that could support multiple information flow pathways.

### 3.3. Mutational Robustness of Evolved Networks

It is not immediately obvious why we witness significantly different levels of computational complexity in networks that achieve the same level of functionality (i.e., perfect fitness given the particular task). Is complexity evolving “for free” because there is no cost of this complexity to the organism? It has been argued previously that complexity can emerge by a “ratchet-like” mechanism in which (selectively neutral) complexity is maintained by sign epistasis [30,31,32]. In those models, complexity can evolve even though there is a cost to this complexity (for example, decreased robustness to mutations). Interestingly, often this cost is offset by increased evolvabilty, as the neutral complexity provides the raw material for evolutionary innovations [33,34].

We will now attempt to assess whether there is indeed a cost of complexity in networks evolved to solve the Block Catch task, expressed in terms of a decreased tolerance to mutations. As previously defined, complexity is the total number of information flow arrows in the flow diagram, using the full-lifetime state recording as the time series. In addition to testing the cost of complexity on Markov Brains, we will here also test Recurrent Neural Networks (RNN) to determine if the observed result is substrate independent. For simplicity, we limit this analysis to only those networks that achieved maximal fitness (when imperfect solvers were included we found the same trends, but the results were less significant).

Figure 9 shows that among perfectly performing brains, more complex brains suffer a greater fitness loss due to mutations compared to simpler brains. This suggests that there is indeed a cost of complexity, both for Markov Brains and RNNs: when information flow is complex, mutations are more likely to disregulate behavior and lead to non-optimal decisions. For Markov Brains, a simple explanation for the correlation might be that more complex Markov Brains have more logic gates (which are encoded by more genes), implying that there are more coding sites in the genomes that can be targeted by mutations. However, this would not explain the similar correlation in RNNs, which have a fixed number of coding sites. We did not test whether the increased complexity has consequences for evolvability, that is, whether the cost of complexity can be offset by an increased likelihood to find beneficial mutations, in particular when environmental circumstances change. Such an investigation is left for future work.

## 4. Discussion

Functional biological and computational networks—whether neuronal networks, gene regulatory networks, protein–protein interaction networks, or digital brains—process information. They sense and receive data and information, relay it to other parts of the network, integrate it with information sensed elsewhere as well as with information previously stored in the network (memories), and then deliver the results of this processing either to other networks or to actuators that implement a decision. Understanding the flow of information in those networks will be key to understand their function, but this task is complicated by the fact that information is not additive, and it is not usually localized within single information carriers (such as neurons, genes, or proteins in biological systems, or registers in digital systems). Here, we introduced tools that allow us to identify correlations between nodes in an information-bearing network and concepts external to the network, or between nodes of the network at different time points, and determine how dispersed (fragmented) this information is. These tools also allow us to follow the flow of information through networks. We use these methods (in particular, the information fragmentation matrix) to find where, in artificial brains evolved to solve particular tasks, information is stored, and how it flows through the brain from sensors downstream to actuators. We find that a single evolutionary process can result in solutions with significantly different computational complexity for the same task, and that these solutions differ drastically in the degree of fragmentation of information. Although solutions with perfectly localized information exist, often more complex solutions evolved in which information was distributed, and encrypted.

We found that the pathways of information flow are not fixed as brains perform a task. In the Block Catch task, we found that complex information flow present during the early part of the task (when the brain ascertains block size and lateral motion) is later replaced by simpler dynamics that encodes the behavior in a more generally localized manner and using less complex information flow (note that it is also possible to see differences between early-stage and late-stage dynamics in the gene regulatory networks that control developmental patterning in *Drosophila* [9]).

This sequence of events (complex information flow during early stages of processing and simple flow in later stages) is reminiscent of the learning process in general (imagine learning to play a new piece on the piano) where, during the initial learning phase, many neurons are involved in complex behavior involving strong feedback loops that link sensory data (visual, auditory, and touch) and muscular execution. As the brain learns the piece, complex processing is gradually replaced by a simpler process in which the result (the playing of the piece) is executed by a smaller set of neurons, resulting in behavior that does not rely nearly as much on senses, and without slow feedback loops.

To understand whether differences in information-flow complexity matter, we tested the robustness of evolved digital brains to mutations. We found that information-flow complexity matters, and not just because more complex information flow is encoded by more genes (as the RNNs that differed in flow complexity do not differ in gene number). We surmise that complex flow (where information is fragmented and/or encrypted) is more fragile, but also speculate that rampant information encryption (which is difficult to undo via mutational events) might provide an advantage in terms of a larger reservoir of integrated information which could be useful for problem-solving, if environments change and rapid adaptation is required. We plan to test these speculations in future work.

The mutational robustness test was inspired by experiments using the Aevol model, found in [34]. Interestingly, although the underlying systems are quite different, we were not only able to replicate the work conceptually, but also found that our results align with those observed in the Aevol study: even when simply solutions are possible, evolution still commonly generates more complex solutions even though they are less mutationally robust.

From a more practical point of view, the modern world has developed an increasing reliance on artificial cognitive systems (i.e., digital brains) mostly in the form of artificial neural networks. These systems are widely used in emerging technology, such as cloud computing and information security, as well as other fields including agriculture, medicine, and manufacturing [35]. As computer power increases, methods employed to develop digital brains and the resulting digital brains themselves become more complex. In fact, digital brains are often so complex that they are effectively black boxes—that is, they can perform tasks to a high degree of accuracy, but the process used to arrive at the results cannot be identified. As vital systems are entrusted to these networks, we must continue to develop better methods aimed at understanding how they “think”, and process both data and information. Fragmentation, and other information-theoretic tools like it, may be useful in unlocking these black boxes by providing new views on the operations of these systems. In addition, understanding information flow in deep neural networks could aid in the development of new algorithms that can detect inefficient resource use (for example, encryption) as well as over-reliance on features resulting in over-fitting.

Fragmentation provides a new angle from which we can observe and analyze complex information networks. With currently available computing power, information fragmentation analysis is limited to small networks. In time, more efficient methods and faster computers will allow us to consider larger systems (e.g., deep neural networks), and even biological networks (such as gene regulatory networks and brains). In the near term, we are hopeful that fragmentation analysis will provide insights into the evolution of cognitive processes, as well as provide a basis for new theory that can be applied to other complex biological networks such as disregulated signal transduction networks that lead to cancer [36,37,38]. In addition, we plan to investigate applying information fragmentation analysis to data sets that are small enough to allow information-based analysis, including recordings of neural circuit activity [39,40] and models of gene regulatory systems [9].

## 5. Methods

### 5.1. Fragmentation, Fragmentation Matrices and Information Flow

#### 5.1.1. Data Collection and Formatting

The calculations for fragmentation F and for fragmentation matrices rely on the equation for shared entropy Equation (Equation 3), which relies on the equation for entropy Equation (Equation 1). The term *P* in these equations refers to a list of probabilities ($p_{i}$) that sum to one, where each $p_{i}$ is the fraction of the time that the system spends in a unique state. For example, suppose we observe the weather over a week, where it was rainy on Sunday and Monday, sunny on Tuesday, rainy again on Wednesday, snowy on Thursday and Friday, and rainy on Saturday. Over the week, there were three states: rainy, sunny, and snowy. Since it was rainy four times, snowy twice and sunny once, the *P* for the week’s weather would be $\left\{\frac{4}{7},\frac{2}{7},\frac{1}{7}\right\}$. (Technically, the $p_{i}$ are the maximum-likelihood estimators of the true probability.)

When we analyze digital brains, the values $p_{i}$ are the joint variables consisting of input values, output values, memory values, or values relating to the state of the world (task), depending on the particular question we are asking. Data is collected during each lifetime of the brain, where a lifetime is one test (in Block Catch, for example, each time the agent is presented with a new block, a new lifetime begins). Each lifetime consists of some number of updates and every update results in a set of state recordings (for input, output, memory, and world), which are collected in lists. Once all evaluations of a brain are complete, the state lists are converted into *P* lists by counting the number of times unique states are observed. We collect world states and input states at time t−1 relative to when we collect output states at time *t*. In order to capture the memory states, we must create two recordings: the t−1 memory recording and the *t* memory recording (i.e., the memory states before each brain update and after each brain update) (Since each t−1 memory state is the *t* memory state from the prior update, we in actuality capture one state recording for memory that has one more element than the recordings for input, output, and world state.)

When we create the fragmentation matrices in Figure 3 and Figure 7 the features (*y*-axis) are world states and the predictors (*x*-axis) are t−1 memory. The flow diagrams Figure 5 and Figure 8 are generated from fragmentation matrices using output joined with *t* memory as features (*y*-axis) and input joined with t−1 memory as predictors (*x*-axis).

In Figure 7 and Figure 8 we show data for the full lifetime and also for the late lifetime. The full-lifetime data uses the entire recording ranges, as indicated in the previous paragraph. For the late-lifetime figures, we trimmed the recorded data so that it was limited to only the last 25% of each lifetime’s recordings. For example, if each lifetime was 20 brain updates and the agent was tested four times, then the full-lifetime data would contain 80 (4×20) state recordings and the late-lifetime data would contain 20 (4×5) state recordings.

#### 5.1.2. Generating Information Fragmentation Matrices

A fragmentation matrix is used to determine where some information is located in a system comprised of individual parts. The rows of the matrix are the features: the variables with information that we would like to locate in the system. The columns of the matrix are the power set (excluding the empty set) of the parts of the system ($S_{m}$). The first step is to calculate the entropy $H(Y_{i})$ of each feature $Y_{i}$. We then fill in the matrix by calculating the shared entropy between each feature and each subset of the system as $I(S_{x}^{\left(i\right)};Y_{j})$, and filling in the cell with the normalized $\frac{I(S_{x}^{\left(i\right)};Y_{j})}{H(Y_{j})}$.

#### 5.1.3. Determining Fragmentation

Given a random variable and a system comprised of individual parts, fragmentation is the smallest subset of the system that shares at least θ information with the variable. Given a fragmentation matrix that contains the feature of interest, we can simply look up the fragmentation by finding the first column where that feature information has a value at least θ.

#### 5.1.4. Visualizing Information Flow

In cases where the predictors and features of an information fragmentation matrix have a temporal relationship (as in this study), we can use the fragmentation matrix to construct an information flow diagram where directed lines show which sets of predictors account for information in each of the features. For each feature, we first identify the *distinct informative sets* (DISs): sets of elements of the predictor system that perfectly predict the feature and are not a proper superset of a smaller DIS. For example, if a set of nodes {A,B,C} perfectly predicts some feature, but the set {A,B} also predicts the feature, {A,B,C} would be ignored because it is a superset of {A,B}. On the other hand, if the set {A,C} predicts the same feature, {A,C} would be identified as a DIS because {A,B} and {A,C} each contain at least one unique element. When there is only one DIS for a given feature, then we know that this set must be the explanation for the feature. On the other hand, if there is more than one DIS for a given feature, there is redundant information flow, and we cannot (without further investigation) be certain which DIS is causally responsible for the feature.

Information flow is visualized as nodes connected by arrows, where nodes represent predictor elements and features while arrows show how information flows from predictor elements to features. If a predictor element is in every DIS for a feature (or there is only one DIS), the arrow is drawn in black. If a predictor element is not in every DIS for a feature (if there is more than one DIS), the arrow is drawn in red. In other words, black arrows indicate necessary connections and red arrows indicate potentially necessary connections. Finally, each link is labeled with the ratio of shared entropy between the predictor element and the feature over the entropy of the feature (i.e., the ratio of entropy in the feature that can be independently accounted for by that predictor element).

If the structure of the system is known (e.g., in this work we know the connectome of Markov Brains, defined by the logic gates that connect nodes) it may be possible to remove some redundant DISs (disambiguate the information flow). This is done by comparing each arrow with the connections present in the brain’s structure to see if the connectome supports the arrow. If a connection (for example, via a logic gate) from a set to a node is not present in the brain’s structure, that set can be removed from the DISs.

### 5.2. Digital Evolution System

In this work we examine two types of artificial cognitive systems (Recurrent Artificial Neural Networks and Markov Brains) in the context of two tasks: *n*-Back and Block Catch. We used the MABE [41] digital evolution research tool to conduct our experiments.

#### 5.2.1. Tasks: *n*-Back

The goal of the *n*-Back task is for the digital brain to correctly remember, and echo, bits that it receives in its sensor. Every update, a new input bit is passed to the brain, which the brain must store and write out at specified time delays. Here, we specified that the output should have temporal delays of 1, 3, 5, 7, and 8, meaning that on each update the brain should output the inputs from 1 update prior, 3 updates prior, 5 updates prior, and so on. Every generation, each agent is tested 25 times on a 33-bit-long string. For this task, each test is termed a “lifetime” (they see 33 bits in their lifetime, after which the brain is reset). During the first eight updates of each lifetime, the brains are not scored (and state information is not collected) so that brains have an opportunity to load the initial input values into their memory. See Figure 6a for a visualization of the *n*-Back task.

For each brain, fitness is calculated as the number of correct answers divided by the total number of answers provided (a number between zero and one). Random guessing will produce a fitness of 0.5 (i.e., half the guesses will be correct by chance). *n*-Back is a simple task in that it requires memory use, but not information integration to achieve perfect fitness. *n*-Back has been used in prior digital evolution work [16].

#### 5.2.2. Tasks: Block Catch

The Active Categorical Perception (ACP) task is a classic task of cognitive science [17,18,26,27,42,43]. Here, we refer to the task by its more colloquial name: ’Block Catch’. In the Block Catch task, an agent is a paddle that can move left and right along the bottom of a two-dimensional rectangular space. Periodically, objects (blocks) are introduced at the top of the space that fall downward at a constant rate. Each block has a size and a lateral movement (either left or right at a constant speed). Agents must either catch or avoid blocks based on the block’s size and lateral motion direction. An agent catches a block if any part of the block intersects any part of the agent when the block has dropped to the same level as the agent. The agent is given a limited sensor scope with four upward facing sensors, separated by a central “blind spot” of width two between the second and third sensors. Agents can move left or right to find, identify, and track the block as it descends. The environment is a rectangular space 32 units high and 20 units wide with periodic boundary conditions in the lateral dimension (i.e., the left and right edges are identified). For each agent, each block type (sizes: 2, 3, and 4 and directions: left, and right) is dropped from each of the 20 possible starting positions. Each individual combination of size, direction, and position is referred to as a “lifetime”. See Figure 6b for a visualization of the Block Catch task.

Given the sensor configuration, agents cannot identify all block sizes or directions in a single update, giving rise to perceptual ambiguity [26] that requires integration of information over time and/or over various sensors. Because Block Catch requires both information integration and memory, it is significantly more challenging than *n*-Back, which only requires memory.

While previous versions of this task [27,43] have focused primarily on the relatively simple decision of catching small blocks and avoiding larger blocks, here we increased the difficulty by making the decision of which objects must be caught or avoided depend not only on size, but also on the direction of movement. Blocks of size 4 moving in either direction and blocks of size 3 moving to the left must be avoided, while blocks of size 2 moving in either direction and blocks of size 3 moving to the right must be caught. Thus, agents must correctly classify both size and direction of each block to achieve perfect fitness. Fitness is a value from 0.0 to 1.0 and is the number of correct catch-or avoid-decisions made over all lifetimes divided by the total number of lifetimes.

### 5.3. Cognitive Systems

In order to illustrate fragmentation, we evolve two types of artificial brains: Recurrent Neural Networks (RNN) and Markov Brains. The action of these brains can be generalized as converting one set of values, $T_{0}$ (input and memory nodes, at time t=0) into a new set of values $T_{1}$ (output and memory nodes, at time t=1) via an internal process defined by the brain. Both brain types use the same number of input and output nodes, as determined by each task, and eight memory nodes. Memory is implemented with nodes that act like additional inputs and outputs. Only where inputs deliver information from the task to the brain and outputs deliver information from the brain to the task, memory values deliver the updated memory values from the prior update. Example schematics of both brains are shown in Figure 10 and described in detail in the following sections.

#### 5.3.1. Recurrent Neural Networks

Here, we implement a simple single-layer Recurrent Neural Network (RNN). Each value in $T_{1}$ is determined by a dedicated summation and threshold function of a weighted value for each $T_{0}$ node and bias value. When the RNN updates, each node’s summation and threshold function adds its particular bias to the summation of each $T_{0}$ value multiplied by the particular weight for that $T_{0}$ node. The function tanh is then applied to the sum, which results in a value in the range [−1,1] which, in turn, is assigned to the appropriate output. Mathematically, if $T_{0}=\left\{X_{1}\cdots X_{n}\right\}$ and $T_{1}=\left\{Y_{1}\cdots X_{m}\right\}$, $w_{ij}$ is the weight matrix, and $c_{j}$ are the biases, then
(14)$$\begin{array}{c} Y_{i}=\tanh\left(\sum_{j=1}^{m}w_{ij}X_{j}+c_{j}\right)\;. \end{array}$$
The structure of the brain is fixed, and a genome is used to determine the node weights and biases via evolution. The weights are limited to the range [−1,1] and the biases are limited to the range [−3,3].

RNN node states are discretized by transforming negative values to zero, and positive values to 1, when copying state values from from $T_{1}$ to $T_{0}$. Prior work indicates that on the Block Catch task, the type of RNN we use here perform better with discretized memory than RNNs with unmodified (continuous-value) memory [44].

#### 5.3.2. Markov Brains

Markov Brains are digital logic networks consisting of a variable number of gates that read from $T_{0}$ nodes and write to $T_{1}$ nodes. Each of the gates takes a variable number of inputs (between 2 and 4), and uses a look-up table that implements a logic gate to generate a variable number of outputs (between 2 and 4). If more than one gate writes to the same $T_{1}$ node, that node takes on the “or” of all of its inputs.

A genome (a string of bytes) determines the number of gates and each gate’s properties using an indirect encoding. Brains are constructed by searching the genome for start codons (a predefined pair of bytes) that indicate the beginning of a gate description (a gene). A section of the genome following each start codon defines the number of inputs and outputs, where these connect and the lookup table for the gate. More details on the Markov Brain architecture and genomic encoding can be found in [15].

### 5.4. Evolutionary Algorithm

At the beginning of each experiment, we initialize genomes with 5000 random bytes. When Markov Brains are being used, we seed the genomes with six start codons. For all experiments, we used a population size of 100 individuals with tournament selection method (tournament size 5) and ran 250 replicates. *n*-Back experiments are run for 40,000 generations. Block Catch experiments are run for 20,000 generations.

Mutation operators are point mutations that randomize the value of a site with a per-site probability of $5\times 10^{-3}$, “insert” mutations that copy a section of the genome of length between 128 and 512 sites into a new location with a per-site probability of $2\times 10^{-5}$, and deletion mutations that delete sections of the genome using the same parameters as the copy mutation. Insert and delete mutations are limited such that the genome is never smaller than 2000 or larger than 20,000 sites. At the end of each run, the line of descent [45] is reconstructed, and agents from the end of the lines of descent are used to generate fragmentation matrices and information flow graphs.

### 5.5. Testing Mutational Robustness

In order to test the mutational robustness of networks, we evaluate the final agent from each experiment’s line of descent to establish a baseline score $B_{0}$ (the fitness). We then produce 100 mutants (using the standard per-offspring mutation rates) and determine the average score of these 100 mutants, 〈B〉. We then calculate the mutational robustness *R* of that networks as:(15)$$\begin{array}{c} R=\frac{\langle B\rangle }{B_{0}}\;. \end{array}$$
In perfect-scoring agents, *R* is simply the average mutant score (as perfect score is B=1.0). Imperfect agents may have *R* that are greater than or less than 1.0; although, in this work, we only show results from perfect scoring agents. 

## Figures and Tables

**Figure 1 entropy-24-00735-f001:**
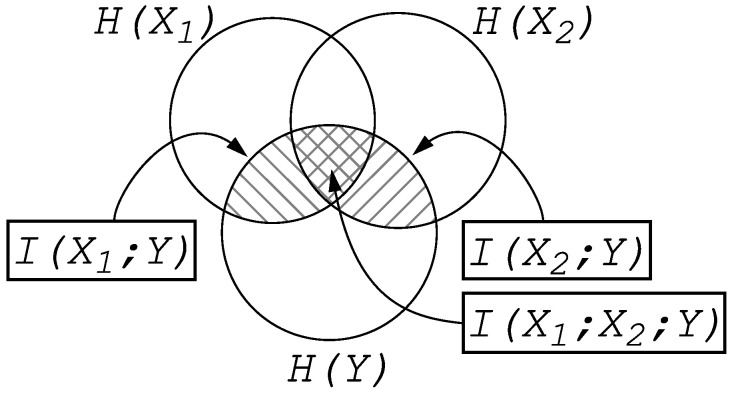
Entropy Venn diagram showing how the information about the joint variable $X=X_{1}X_{2}$ stored in *Y* is distributed across the subsystems $X_{1}$ and $X_{2}$. The information $I(X_{1};Y)$ shared between $X_{1}$ and *Y* is indicated by righthatching, while the information $I(X_{2};Y)$ is shown with lefthatching. As $X_{1}$ and $X_{2}$ can share entropy, the sum of $I(X_{1};Y)$ and $I(X_{2};Y)$ double counts any information shared between all three: $I(X_{1};X_{2};Y)$ (crosshatched). Because information shared between three (or more) parties can be negative, the sum $I\left(X_{1};Y\right)+I\left(X_{2};Y\right)$ can be larger or smaller than I(X;Y).

**Figure 2 entropy-24-00735-f002:**
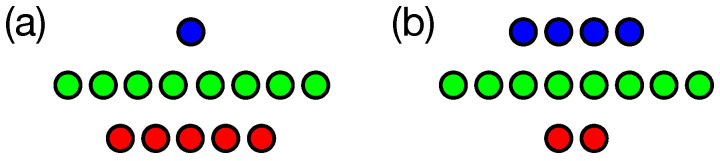
Node configuration for Markov Brains and Recurrent Neural Networks. (**a**) Networks for the *n*-Back task have a single input node (blue), eight memory nodes (green) and five output nodes (red) to report on prior inputs. (**b**) Networks for the Block Catch task have four “retinal” or sensor (input) nodes, eight memory nodes, and two motor (output) nodes that allow the agent to move left or right.

**Figure 3 entropy-24-00735-f003:**
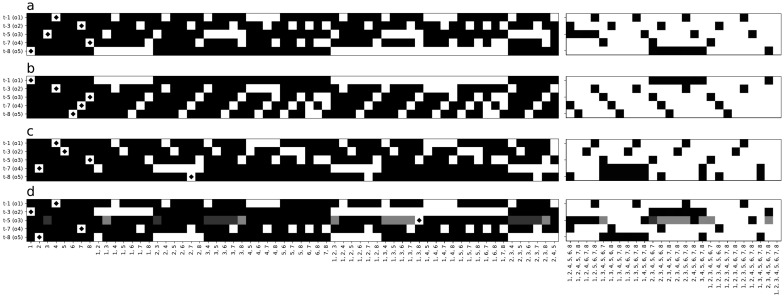
Fragmentation matrices for the *n*-Back task. Matrices from four Markov Brains evolved on the *n*-Back task that evolved perfect performance, shown as (**a**–**d**). The features labeling the rows of the matrix are the expected outputs on the current update, while sets (labelling columns) are combinations of the brain’s eight memory values $m_{1}\cdots m_{8}$. The amount of information between each feature and each set is indicated by gray-scale, where white squares indicate perfect correlation, and gray to black represents successively less correlation. The black diamond within a white square indicates the smallest distinct informative set (DIS) that predicts each feature. A portion of each matrix containing sets of intermediate size is not shown to save space.

**Figure 4 entropy-24-00735-f004:**
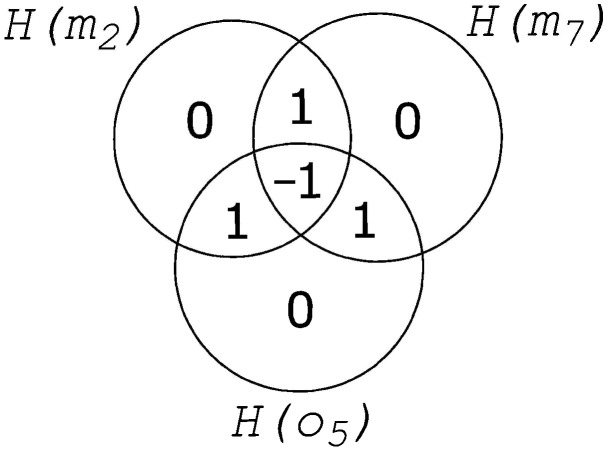
Entropy Venn diagram for element {$o_{5},m_{2},m_{7}$} of the fragmentation matrix shown in Figure 3c. As $o_{5}\left(t\right)=m_{2}\left(t-1\right)⊗m_{7}\left(t-1\right)$ (⊗ is the XOR operator), information about $o_{5}$ is perfectly encrypted so that each of the nodes $m_{2}$ and $m_{7}$ reveal *no* information about $o_{5}$. Because this Venn diagram is symmetric, it is arbitrary which variable is called the sender, the receiver, or the key.

**Figure 5 entropy-24-00735-f005:**
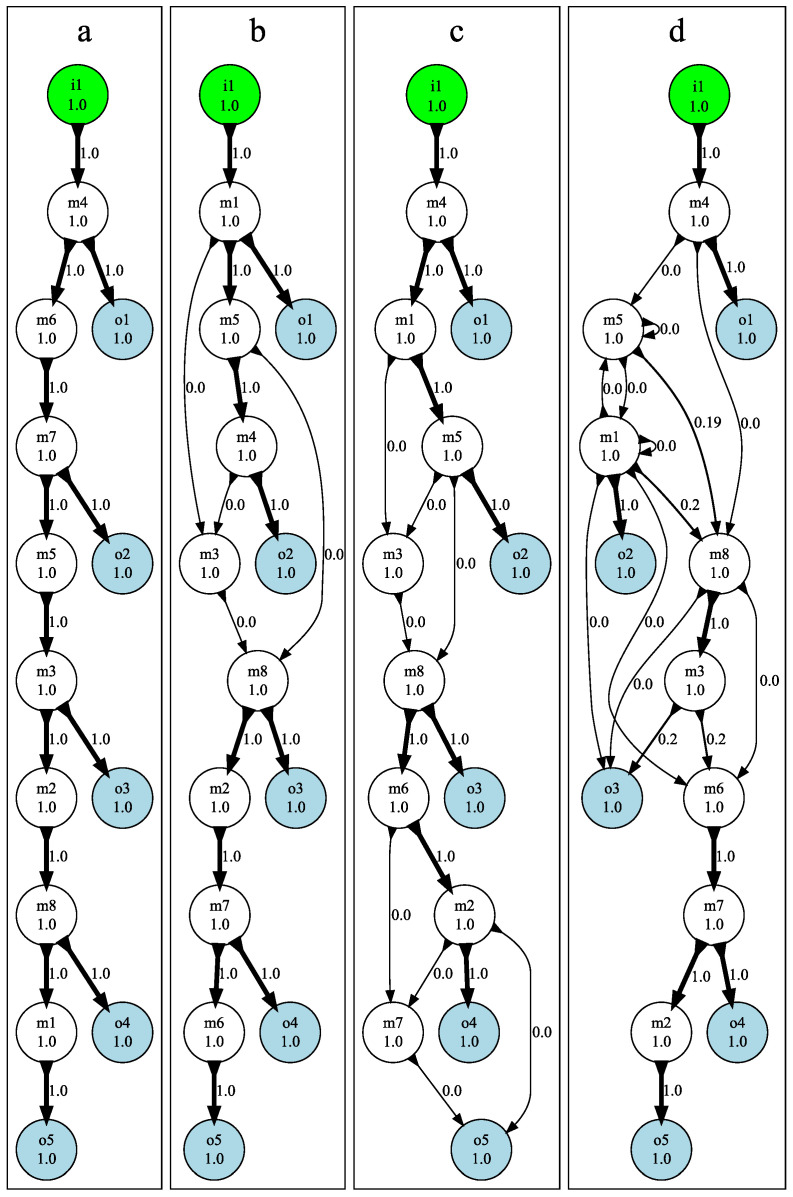
Information flow through nodes of the Markov Brains evolved to solve the *n*-Back task. Diagrams (**a**–**d**) correspond to the fragmentation matrices shown in Figure 3a–d. Input node $i_{1}$ is in green, output neurons $o_{k}$ are blue, and memory neurons $m_{k}$ are white. The numbers within the nodes are the entropy of that node throughout a trial (as the inputs are random, each node has one bit of entropy). The arrows going into each node represent the connections necessary to account for the total entropy in that node. The labels accompanying each arrow and the arrows’ widths both indicate the proportion of the entropy in the downstream node that can be accounted for by each arrow alone, but because information is distributed and not additive, the sum of informations often does not equal the entropy of the downstream node. Memory nodes with zero entropy are not shown to simplify the graphs (all brains have eight memory nodes). In this configuration, *n*-Back agents were required to report on the outputs correspondent to t−1, t−3, t−5, t−7 and t−8, where *t* is the current time.

**Figure 6 entropy-24-00735-f006:**
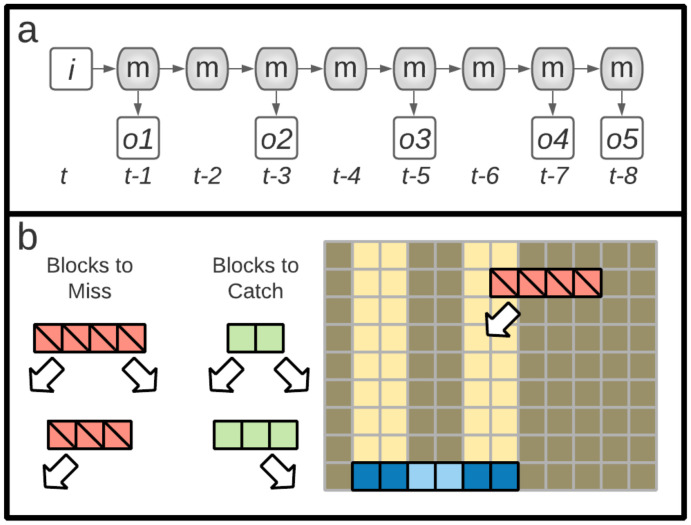
Depiction of the two tasks used. (**a**) In the *n*-Back task, successive bits provided to the agent at input *i* and must pass though various portions of the memory *m* and delivered to outputs at later times, such that the outputs $o_{1}$, $o_{2}$, $o_{3}$, $o_{4}$, and $o_{5}$ at a given time *t* provide the input state from prior time points t−1, t−3, t−5, t−7, and t−8, respectively. (**b**) In the Block Catch task, blocks of various sizes with left or right lateral motion are dropped. Some blocks must be avoided (those shown in red) while other blocks (shown in green) are to be caught. The right portion of (**b**) shows a subsection of the environment at a particular moment, with a left-falling size-4 block (red). The agent is depicted in blue, with the sensors in dark blue and the “blind spot” in light blue. As currently positioned, only the rightmost sensor of the agent would be activated. Here, the agent should miss the block. The agent “catches” a block if any part of the block intersects any part of the agent.

**Figure 7 entropy-24-00735-f007:**
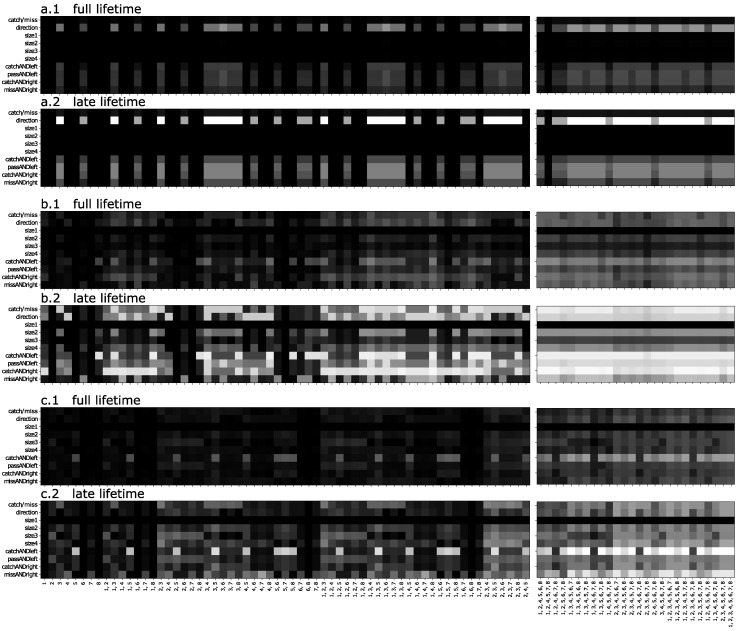
Fragmentation matrices for three Markov Brains evolved to perfect performance on the Block Catch task. For each brain two fragmentation matrices are shown, the first using the state information from the full lifetime (all 120 conditions for 31 updates), and the other only the late lifetime, that is, the last 25% of updates (all 120 conditions). The features (labeling the rows) represent various salient features of the world state. The columns are combinations (sets) of the brain’s 8 memory nodes $m_{1}\cdots m_{8}$. The amount of information between each feature and each memory set is indicated by gray-scale, where white squares indicate perfect correlation, and gray to black represents successively less correlation. A portion of each matrix containing sets of intermediate size is not shown to save space. (**a**) Full-lifetime fragmentation matrix of a simple brain (1), same brain, late-lifetime fragmentation matrix (2); (**b**) full-lifetime and late-lifetime fragmentation matrices for an intermediate-complexity brain (1 and 2, respectively); (**c**) full-lifetime and late-lifetime fragmentation matrices for a complex brain (1 and 2, respectively).

**Figure 8 entropy-24-00735-f008:**
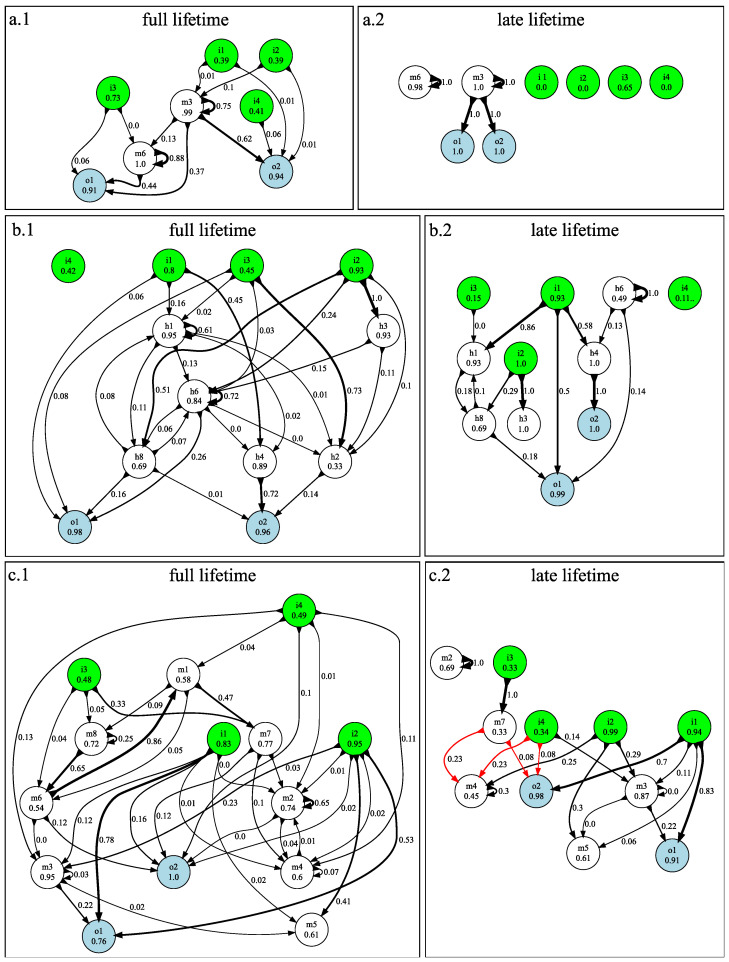
Full-lifetime (**a.1**,**b.1**,**c.1**) and late-lifetime (**a.2**,**b.2**,**c.2**) information flow diagrams for the Block Catch task, for the three brains shown in Figure 7. Green, white, and blue nodes indicate inputs (*i*), memory (*m*), and output (*o*) nodes respectively. The numbers in the nodes indicate the entropy (in bits) in that node. The labels accompanying each connecting link and the link’s width both indicate the proportion of the entropy in the downstream node that can be accounted for by that link. The links rendered in black going into each node represent the connections necessary to account for the total entropy in that node. Red links indicate connections that may (but do not necessarily) account for downstream information (indicating redundant predictive sets). Memory nodes with zero entropy are not shown to simplify the figures (all brains have eight memory nodes). Figure labels correspond to results shown in Figure 7.

**Figure 9 entropy-24-00735-f009:**
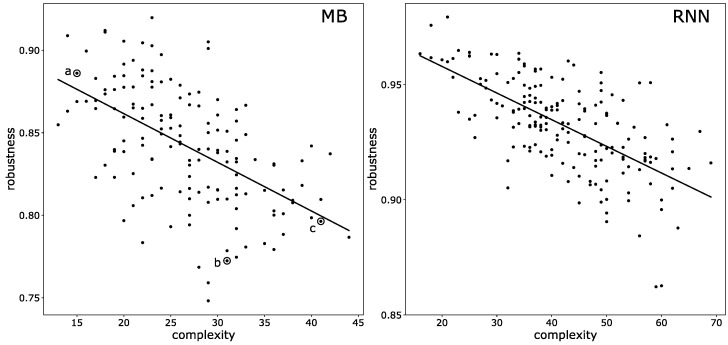
Mutational robustness (average degradation of performance) vs. flow-complexity (number of informative arrows in the information flow diagram), for Markov Brains (left panel) and RNNs (right panel). In the left panel, three dots are circled and annotated (a–c) to indicate values generated by the three networks shown in Figure 7 and Figure 8. Black solid lines indicate a line of best linear fit.

**Figure 10 entropy-24-00735-f010:**
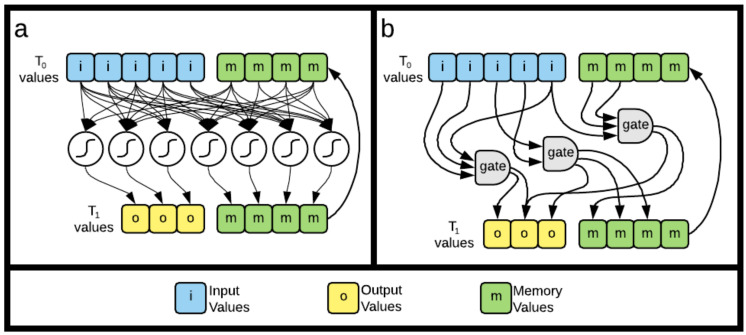
Depiction of the two cognitive systems used in this work. Both brain types have the same general structure, which consists of a “before” state, $T_{0}$ and an “after” state, $T_{1}$. The $T_{0}$ state is made up of inputs and prior memory, while the $T_{1}$ state is made up of outputs and updated memory. (**a**) shows the structure of the RNNs where data flows from $T_{0}$ (input and prior memory) through summation and threshold nodes to $T_{1}$ (outputs and updated memory). (**b**) shows the structure of the Markov Brains, where information flows from $T_{0}$, through genetically encoded logic gates, to $T_{1}$.

## Data Availability

The code used to generate the results in this document as well as replication instructions can be found at https://github.com/cliff-bohm/Fragmentation_Replication_Instructions (accessed on 20 May 2022).

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
