# Peer review of "Information Fragmentation, Encryption and Information Flow in Complex Biological Networks"

_entropy, 2022, doi:10.3390/e24050735_

Round 1

Reviewer 1 Report

In this paper, the author propose a new information-theoretic technique, termed "information fragmentation analysis", for studying how information is distributed in multivariate systems. They demonstrate this technique on agents evolved to carry out two tasks: a memory recall task and a block-size recognition task.

The paper is well-written and the scientific content is acceptable (though nothing beyond this).

I recommend the work for publication, after the following concerns have been addressed:

1) It is not entirely clear why yet another information-theoretic technique for analyzing multivariate systems is needed. In fact, dozens of such methods already exist in fields like computational neuroscience (e.g., functional connectivity methods), computational biology (e.g., finding informative subsets of genes), machine learning (e.g., information-theoretic feature selection), information theory (e.g., reconstructability analysis, partial information decomposition, &c.), and various other fields. It is a serious issue that none of these methods are discussed, and most are not even cited.

In my opinion, the benefit of the authors contribution is largely unclear, unless the results produced by fragmentation analysis are compared to results produced by other methods (such as functional connectivity methods based on transfer entropy), although perhaps this can be done in a future paper.

2) I am surprised that the authors did not cite
* R. D. Beer, P. L. Williams, Information Processing and Dynamics in Minimally Cognitive Agents, Cognitive Science 39 (2015)
which analyzed a variant of the block task from an information-theoretic point of view, and with very similar motivations. It should be discussed how the insights produced by fragmentation analysis go beyond what was done in the work by Beer & Williams.

Minor comments:

  • Around page 2, it is not clear why the authors do not use the more common term "mutual information" rather than "shared entropy". Also, the definition of "shared entropy" for three variables needs to be unpacked, and relevant literature cited. If I understand correctly, this is what is typically called multi-information or interaction information. Literature on the growing field of partial information decomposition should be cited here.
  • Related to the above, what the authors term "encryption" is often termed "synergy" in the information theory literature.
  • Typo: line 46, "Notice, that", line 316, "s"->"a"

Reviewer 2 Report

The paper from C. Bohm et al proposes a information measure that grasps how distributed is the information content about the random variable Y among the different sets of the power set of the values that another random variable X can take.

I think this is an interesting proposal and has potential to be useful in order to disentangle information processing in distributed systems. However, the paper is written in a very confusing way, and it needs to improve its presentation in order to properly grasp the relevance and the depth of the proposed analysis --which, as said, I think it has potential. As it is now, the reading the paper is a tour-de-force trying to infer what the authors want to say. Since I think the theoretical part must be clearly improved, I will not comment the numerical experiments, as I need more understanding to judge them.

My points are:

i) The introduction contains only 1 citation, which is odd if one thinks about some strong claims that are made there.

ii) Page 3: is not clear what is a random variable with m elements or parts. Is the random variable taking values in a discrete set made of m elements? According to a given probability distribution?

iii)Equation (3) should be a strict inequality

iv)Usually what authors call H(X:Y) is referred in the literature as I(X:Y) --take the reference of Cover and Thomas, 2001, for example. 

v) Distinct informative set: please, define it before equation (8)

iv) Please, better explain the role of the role of the parameter θ. Is it a threshold that we impose externally?

vi) How can you ensure unicity of the DIS if the parameter θ is a variable? I guess different choices may lead to different DIS configuration for the same problem.

vii) Equation (9): What is a "feature"? A random variable? The random variable Y? Please describe it accurately and explain the map and the potential change of language, if required

viii) How does a network enters the formalism? The nodes are the states? Or configuration of states of the nodes?

I will be happy to read a revised version of the paper, because I think the approach is valuable.

Round 2

Reviewer 1 Report

-